

# Apparent Friction Coefficient Used for Flow Calculation in Straight Compound Channels With Trees On Floodplains

Adam P. Kozioł[1], Adam Kiczko[1], Marcin Krukowski[1], Elżbieta Kubrak[1], Janusz Kubrak[1], Grzegorz Majewski[1], Andrzej Brandyk[1]

[1]Institute of Environmental Engineering, Warsaw University of Life Sciences, Warsaw, Poland

*Correspondence to*: Adam P. Kozioł (adam_koziol@sggw.edu.pl)

**Abstract.** The interaction of water streams in channels with a complex cross-section, involving the exchange of water mass and momentum between slowly-flowing water in the floodplains and fast water in the main channel, significantly depends on the diversification of the surface roughness between the main channel and floodplains. Additionally, trees strongly increase flow resistance on floodplains, but also significantly in the main channel by intensifying the interaction process. As a result, the water velocity and the discharge capacity of both parts of the channel decrease and at the same time, affecting the flow conditions in the main channel. The results of laboratory experiments were used to determine the effect of floodplain trees on the discharge capacity of the channel with diversified roughness. The reduction in velocity of the main channel caused by the stream interactions is described with the apparent friction coefficients introduced at the boundary between the main channel and the floodplain. The values of resistance coefficients and their changes as a result of the significant influence of trees on the interaction process were determined for various roughness's of the main channel bottom.

## 1 Introduction

As described in Kubrak et al. (2019), the kinematic structure of the stream in the channel with a compound cross-section can be characterized with sufficient accuracy, as the distribution of the depth averaged velocity in the cross-section. The lateral velocity distribution depends on the channel shape in the plane, the shape of the cross-section, the roughness of the bottom compound channel, and flow resistance caused by the turbulent exchange of water masses and momentum between faster flowing water in the main channel, and slower in the floodplains. The very diversification of bottom surface roughness intensifies the momentum exchange process along with the creating of vortex structures in the transition area between the flood plains and the main channel ("kinematic effect", Zheleznakov, 1971, nowadays is described as the streams interaction). The intensification of the process of creating of vortices and secondary flows in the main channel resulted in a decrease in water velocity, changes in the turbulent flow structure and affects the capacity of the channel with a complex cross-section (Shiono and Knight,1991; Tominaga and Nezu, 1991; Bousmar and Zech, 1999; Rowiński et al.,1998 and 2002; Van Prooijen et al., 2000; Czernuszenko et al. 2007; Kozioł and Kubrak, 2015). Trees from the floodplain further strongly increase the interaction between the main channel and the floodplain. Floodplain trees additionally cause a significant increase in flow resistance, a





reduction in water velocity, a reduction in the capacity of both parts of the riverbed and especially a significant change in the turbulent flow structure (Kozioł, 2008, 2011, 2012, 2013, 2015 and 2019, Mazurczyk, 2007). The results of laboratory experiments (Kozioł, 2013) showed that trees on the floodplains did not result in significant changes in values of relative turbulence intensity in the whole compound channel, but they did result in significant changes in the vertical distributions of the relative turbulence intensities in all three directions on the floodplains and over the bottom of the main channel.

The phenomenon of interaction emerging in the transition region between the main channel and the floodplain is described by separating the two streams, most often with vertical lines, on which the apparent shear stresses were assigned. The concept of the apparent tangential stresses at the division boundaries of the channel compound cross-sections was introduced by Wright and Carstens (1970). Since the 1980s, in accordance with the concept of apparent shear stress, a number of formulas have been introduced based on hydraulic experiments in channels to calculate flow resistance due to momentum transfer between the

main channel and the floodplain (Myers, 1987; Wormleaton et al., 1982; Knight and Demetriou, 1983; Prinos and Townsend, 1984; Christodoulou, 1992). An overview of these formulas can be found in Moreta and Martin-Vide (2010).

Laboratory tests allow the determination of apparent shear stresses, which enable the determination of the values of dimensionless resistance coefficients used to calculate the average velocity in the steady uniform flow in the main channel of the compound cross-section, according to the Darcy-Weisbach formula:

$$v_m = \sqrt{\frac{8gR_mS_o}{f_m}},$$    (1)

where: $v_m$ - average flow velocity in the main channel, $g$ - gravitational acceleration, $f_m$ - resistance coefficient for the main channel cross-section, calculated for the wetted perimeter, accounting for the length of the cross-section division plane, side slopes and the bottom of the main channel, $R_m$ - hydraulic radius of the main channel cross-section, $S_o$ - longitudinal channel slope.

The flow resistance coefficients at the division planes of the compound cross-section, calculated on the basis of apparent shear stresses, depend on the channel parameters given by Nuding (1998), but in the case of trees on the floodplain they also depend on additional parameters such as: $d$ - tree diameter, $A_v/A$ - the degree of cover of the cross-sectional area of the channel by trees, $a_x$ and $a_y$ - spacing of trees in the longitudinal and transverse directions. Then the flow resistance coefficients depend on the following channel parameters (Fig. 1):

$$f_a = f\left(f_{mb}; \frac{H}{h_f}; \frac{b_m}{b_f}; 1:m; \frac{k_{mb}}{k_{fb}}; d; \frac{A_v}{A}; a_x \text{ and } a_y\right),$$    (2)

where: $f_a$ - apparent coefficient of resistance at the boundary between the main channel and floodplain area, $f_{mb}$ - resistance coefficient of the main channel bottom, $f_{ms}$ - resistance coefficient of the main channel side slopes, $f_m$ - resistance coefficient in the main channel, $f_{fb}$ - resistance factor of the bottom of the floodplain, $H$ - water depth in the main channel, $h_f$ - water depth on the floodplain, $b_m$ - bottom width of the main channel, $b_f$ - floodplain width, 1:$m$ - aspect of the side slope of the main





channel and floodplains, $k_{mb}$ - absolute surface roughness of the main channel, $k_{ms}$ - absolute roughness of the main channel side slopes, $k_{fb}$ - absolute surface roughness of the floodplain, $k_{fs}$ - absolute roughness of the floodplain side slopes.

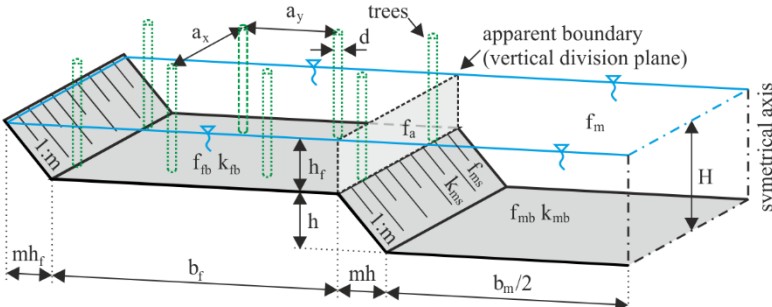

**Figure 1: Symbols used for dimensions of the compound cross-section of the channel.**


Bretschneider and Özbek (1997) used measurements of average water velocity in the main channel, and apparent tangential stresses at the division boundary of the cross-section on large-scale hydraulic models, to determine the apparent resistance coefficients on vertical division lines. Kubrak et al. (2019) performed research on a small-scale hydraulic model and the aim of their research was to explain how the surface roughness of the main channel and floodplains affects the values of apparent

resistance coefficients. The main goal of this unique work was to determine the influence of floodplain trees on the value of the apparent resistance coefficient in a compound channel for various roughnesses of the main channel bottom. The results of measurements from previous experimental studies in compound channels on the flow capacity and the turbulence structure were used to write this manuscript (Kozioł, 1999, 2012, 2013, 2019; Kubrak et al., 2019).

**2 Study on Discharge Capacity of Channel with Compound Cross-Section**

Study on the capacity of the channel with the compound cross-section was carried out in the hydraulic laboratory of the Department of Water Engineering of the Warsaw University of Life Sciences. A straight open channel (16 m long and 2.10 m wide) with a symmetrically trapezoidal cross section was used for the laboratory variants (Figs. 1, 2, 3 and Kozioł, 2013, Fig. 1). The bottom of the main channel and symmetrical floodplains in the cross-section were horizontal. The main channel width was 30 cm, and the floodplain width was 60 cm. The sloping banks were inclined at a slope of 1:1. The channel bed slope of

the channel was 0.5 ‰. The channel model was supplied by five pumps with a total discharge of 0.50 m³/s in a closed water cycle. To measure the flow rate in the channel, a calibrated circular measuring overflow with a diameter of 540 mm was used. In the initial section of the channel, a row of 0.30 m long PVC pipes was laid, calming the flow, and directing water into the model. A uniform and steady flow was used in every case. The water surface was kept parallel to the bed during the experiments. The water surface slope was measured by recording the pressure differences between readings of piezometers

located along the centerline of the channel bed at the distances 4 and 12 m from the channel entrance. Measured values were





the flow rate in the channel, the water depths in the main channel and on the floodplains, the flow velocity at the cross-section points and the water temperature. The water depths were measured with a pin gauge having an accuracy of 0.1 mm. Before general measurements were started, some trial velocity measurements were performed in a few cross sections at the distances 4, 8 and 12 m from the channel entrance. The Reynolds number was calculated, and it was checked to be sufficiently large to

create the state characterized by local isotropy and homogeneity and associated universal behavior of statistical properties. The cross section halfway down the channel length was selected for velocity measurements (Figs. 1 and 3).

Two devices were used to measure the components of the flow velocity: an electromagnetic PEMS probe and an acoustic Doppler velocity meter (ADV). At the beginning, the electrostatic PEMS probe was used, and then from the newly purchased acoustic ADV probe. The description of the electrostatic PEMS probe, the measurement technique and the method of

determining the required length of the velocity measurements time series are presented in the work of Kubrak et al. (2019) and the ADV probe in the works of Kozioł (2012, 2019). The velocity measurements at a point by the PEMS probe were carried out in 77 measurement verticals and in nearly 500 cross-section points (Kubrak et al., 2019), and by the second probe at 250 points at 23 verticals — 6 on each floodplain and 11 in the main channel (Kozioł, 2012, 2019). The probes were mounted on a sliding measuring carriage. The differential pressure gauge and the probes were connected to a computer measurement logger.

The results of measurements from two experimental studies in compound channels on the flow capacity (Kozioł, 1999; Kubrak et al.,2019) and turbulence characteristics of the water stream (Kozioł, 1999, 2012, 2019) were used to write this manuscript. Diversification of the surface roughness in the channel was obtained by painting the concrete of a blurred surface with paint (called a smooth surface, Fig. 2a), or by applying a terrazzo layer with a grain diameter of 6–12 mm (called a rough surface, Fig. 2b-f).

Compound channel capacity experiments were presented for six variants and for three roughness values of floodplains (Fig. 2).

1. In the first variant (W 1.0, Fig. 2a), the surface of the channel bed was smooth (Manning's roughness coefficient $n = 0.011$ m$^{-1/3}$s, $k_{mb} = k_{ms} = k_{fb} = k_{fs} = 0.00005$ m).

2. In the second variant (W 2.0, Figs. 2b and 3a), the surface of the main channel bed was smooth ($k_{mb} = k_{ms} = 0.00005$ m) and

made of concrete, whereas the floodplains were covered by cement mortar composed with terrazzo ($k_{mb} = k_{ms} = 0.00005$ m, $k_{fb} = k_{fs} = 0.0089$ m). In the variants W 1.0 and W 2.0, values of the absolute roughness of flume surfaces were determined from the distribution of mean velocity in the region where it satisfies the log-law (Nezu and Nakagawa, 1993).

3. In the third variant (W 3.0, Fig. 2c), the surface of the main channel bed was smooth (Manning's roughness coefficient $n = 0.011$ m$^{-1/3}$ s) and made of concrete, whereas the floodplains and all sloping banks were covered by cement mortar composed

with terrazzo. The surfaces of the floodplains were covered by the same cement mortar with the terrazzo. However, studies showed that calculating the Manning's roughness coefficient and absolute roughness of both surfaces had been different. The Manning's roughness coefficient for the rough surfaces of the floodplain equalled about $n = 0.018$ m$^{-1/3}$s for the left and 0.025 m$^{-1/3}$s for the right area. The values of average Manning's coefficient and absolute roughness of the channel surface were determined from the Manning equation and the Colebrook-White equation on the basis of the average velocity values





of the flow measured in those parts of the channel. The obtained roughness amounted to $k_s$ = 0.00005 m for the smooth

surfaces, $k_s$ = 0.0074 m for the rough surface of the left floodplain, and $k_s$ = 0.0124 m for the rough surface of the right

floodplain.

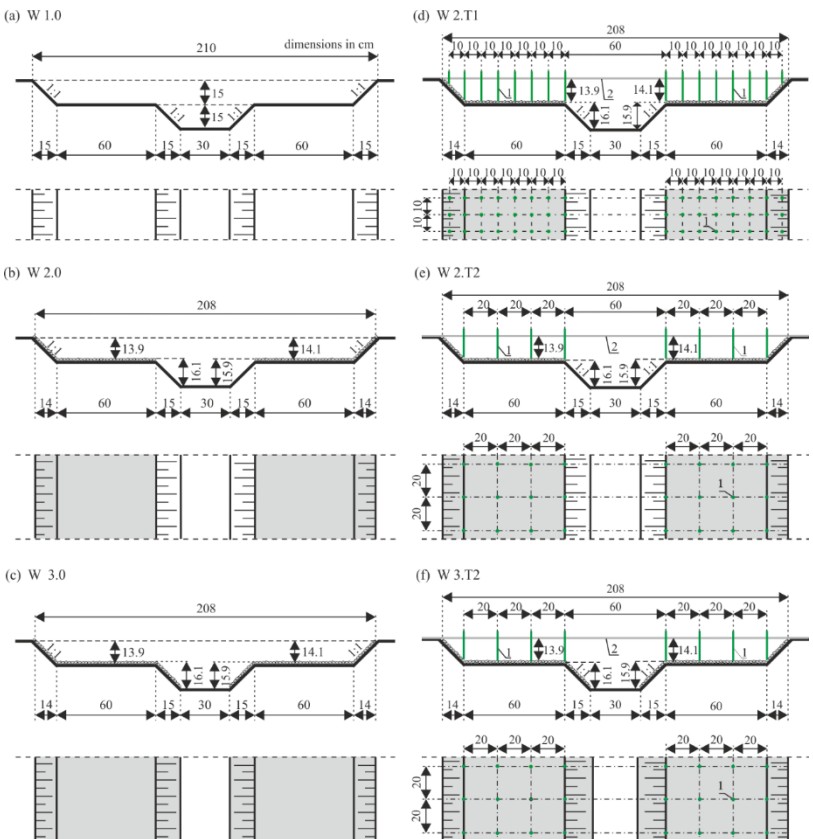

**Figure 2: Cross-section schema of the channel in the analyzed variants (dimensions in cm).**

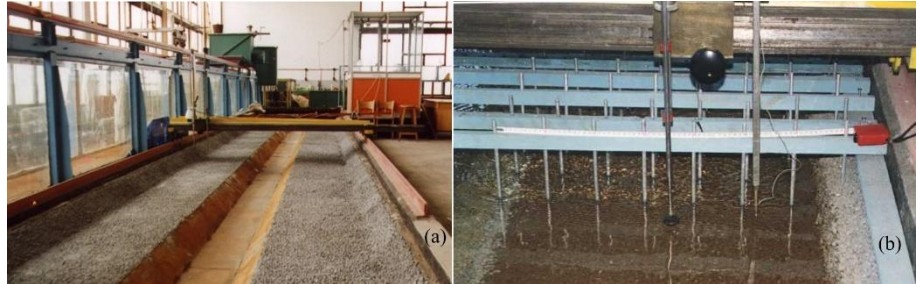

**Figure 3: View of the channel model in variants: (a) W 2.0 and (b) W 2.T1.**





4. In the fourth variant (W 2.T1, Figs. 2d and 3b), the covering of the floodplains was the same as in the second variant, but
emergent vegetation (trees) growing on the floodplains were modelled by aluminium pipes of 0.8-cm diameter, placed with
both longitudinal and lateral spacings of 10 cm. There were sixteen pipes in each of 161 cross sections. The treetops were
emergent, and the pipes were not subject to any elastic strains caused by overflowing water.

5. In the fifth variant (W 2.T2, Fig. 2e), the covering of the floodplains was the same as in the variants W 2.0 and W 2.T1, but
emergent vegetation (trees) growing on the floodplains were modelled by half as much, placed with both longitudinal and
lateral spacings of 20 cm. There were eight pipes in each of 80 cross sections.

6. In the sixth variant (W 3.T2, Fig. 2f), the covering of the floodplains was the same as in the variant W 3.0 (Fig. 2c), but
emergent vegetation (trees) growing on the floodplains were modelled as in the variant W 2.T2 with spacings of 20x20 cm.

The list of experiments carried out during the experiments, measured flow rates in the main channel and the adjacent
floodplains, is summarized in Table 1.

**Table 1: Hydraulic parameters of experiments.**

| Parameter | Variant | | | | | | | |
|---|---|---|---|---|---|---|---|---|
| | 1.0.14 | 2.0.7 | 2.T1 | 2.T2 | 3.A1 | 3.A2 | 3.T2.A1 | 3.T2.A2 |
| Discharge $Q$ [m³s⁻¹] | 0.1317 | 0.0808 | 0.0499 | 0.0613 | 0.0952 | 0.0811 | 0.0657 | 0.0589 |
| Discharge in the Main Channel $Q_m$ [m³s⁻¹] | 0.0703 | 0.0481 | 0.0401 | 0.0426 | 0.0500 | 0.0457 | 0.0386 | 0.364 |
| Discharge in the Left Floodplain $Q_{fl}$ [m³s⁻¹] | 0.0323 | 0.0150 | 0.0051 | 0.0095 | 0.0226 | 0,0180 | 0.0135 | 0.114 |
| Water depth $H$ [m] | 0.253 | 0.251 | 0.253 | 0.256 | 0.283 | 0.264 | 0.280 | 0.263 |
| Water depth in the floodplain $h_f$ [m] | 0.103 | 0.091 | 0.093 | 0.096 | 0.123 | 0.104 | 0.12 | 0.103 |
| Reynolds Numbers in the Main Channel $R_{em}$ | 292,497 | 202,824 | 149,136 | 157,071 | 160,460 | 149,521 | 122,133 | 119,100 |
| Reynolds Numbers on the Left Floodplain $Re_{fl}$ | 167,410 | 79,827 | 23,861 | 44,347 | 92,468 | 74,600 | 54,350 | 47,254 |
| Type of surface | smooth channel | smooth main channel and rough floodplains | | | rough floodplains and sloping banks of the main channel, with smooth bottom of the main channel | | | |
| The arrangement of trees [cm] | - | - | 10x10 | 20x20 | - | - | 20x20 | 20x20 |
| Percentage reduction in flow $dQ_i$ [%] ($i$ – variant no) | | $dQ_{1.0.14-2.0.7}$ | $dQ_{2.0.7-2.T1}$ | $dQ_{2.0.7-2.T2}$ | | | $dQ_{3.A1-3.T2.A1}$ | $dQ_{3.A2-3.T2.A2}$ |
| | | -38.6 | -38.2 | -24.1 | | | -31.0 | -27.4 |

On the basis of spot velocity measurements, it was possible to plot lines of constant velocities (isovels) in the cross-sections
of the channel for all analyzed variants. Examples of isovels in the cross-section of the channel at similar depth of flow ($H \approx$
0.25 m) for variants W 1.0, W 2.0 and variants W 2.T1, W 2.T2 with trees on the floodplains are shown in Fig. 4. Figure 5





presents isovels in the cross-section of the channel with smooth surface of the bottom of the main channel, rough surface of the sides slopes of the main channel and floodplains, for variant W 3.0 and variant W 3.T2 with trees on the floodplains ($H = 0.28$ m).


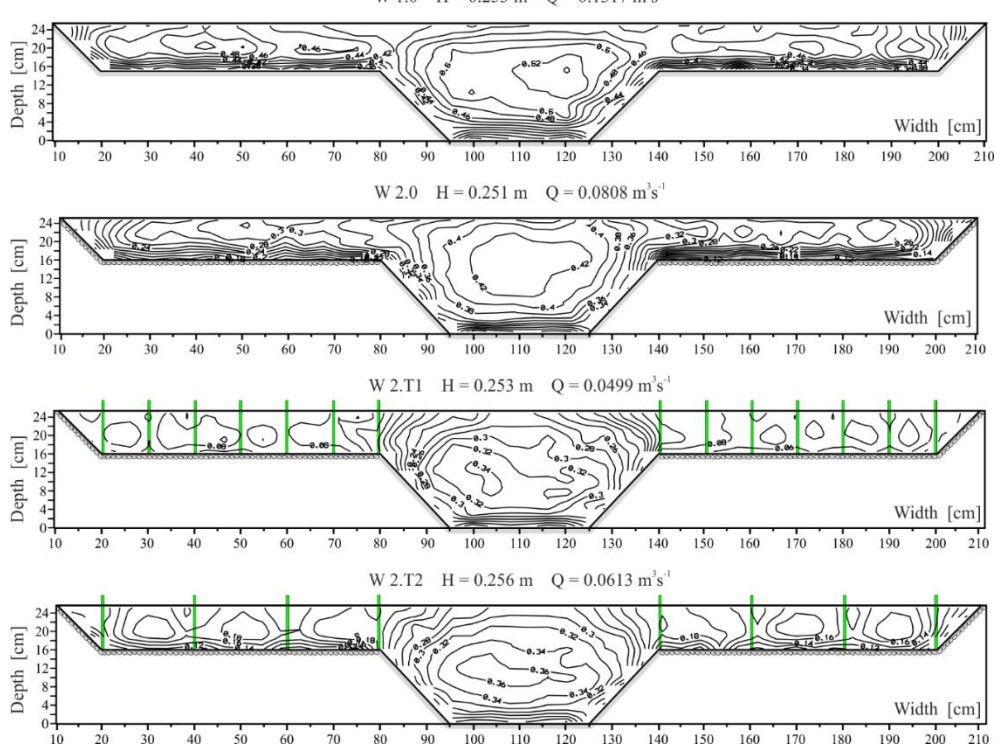

**Figure 4: Isovels in the cross-section of the channel at similar depth for variants W 1.0, W 2.0 and variants W 2.T1, W 2.T2 with trees on the floodplains.**

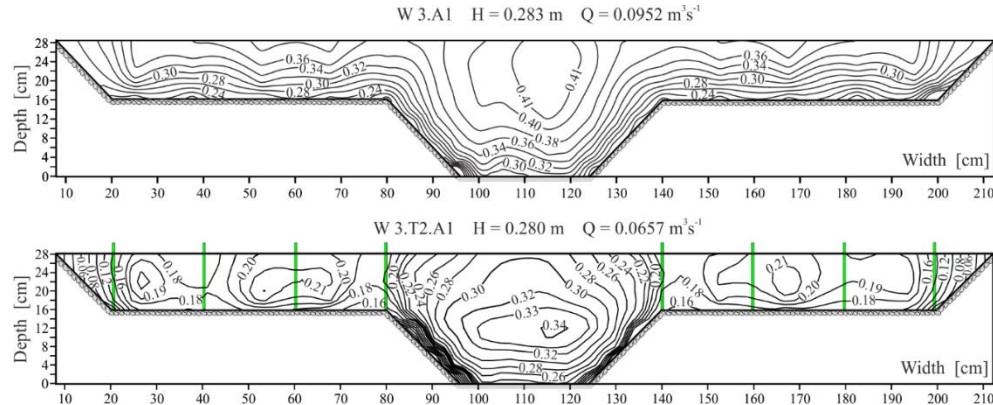


**Figure 5: Isovels in the cross-section of the channel at similar depth for variant W 3.0 and variant W 2.T2 with trees on the floodplains.**





## 3 Resistance Coefficients in the Main Channel

In variants with trees on floodplains, as in variants without trees (Kubrak et. al, 2019), the values of dimensionless resistance
coefficients in the main channel with different bottom and slope roughness, as well as resistance factors $f_a$ in the plane of
distribution of the cross-section of the channel, were calculated using the Einstein method (Einstein, 1934). According to this
method, for each surface roughness along the perimeter of the cross-section the flow area can be found, in which this roughness
shapes the flow conditions. The areas were determined using the graph of isovels in the cross-section of the main channel (Fig.
6). The division of the cross-section into these areas is carried out with lines perpendicular to the isovels, starting from the
wetted perimeter points, separating the perimeter into sections with different roughness (Fig. 6). This technique for determining
division lines assumes that they are free from the shear stress, and forces are not transferred between the separated areas.

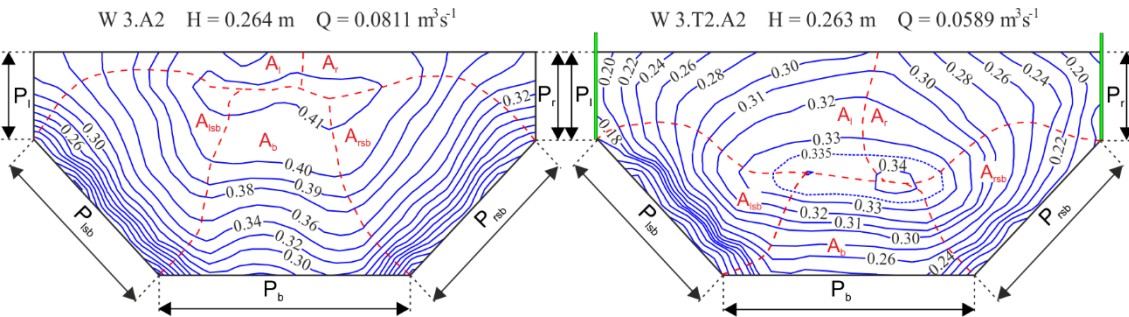

**Figure 6: Surface areas of stream cross-sections $A_i$ in which the flow conditions are shaped under the influence of a constant**
**roughness over the length of the wetted perimeter $P_i$.**

According to the Einstein method, the average flow velocity in each of these sections $A_i$ is equal to the average velocity across
the entire main channel cross-section, i.e., $v_i = v_m$. Expressing the velocity with the Darcy-Weisbach equation with this
condition, the following dependence can be obtained (Kubrak et. al, 2019):

$$\sqrt{\frac{8gR_iJ}{f_i}} = \sqrt{\frac{8gR_mJ}{f_m}} \Longrightarrow f_i = f_m \frac{R_i}{R_m}, \tag{3}$$

where: $f_m$ denotes the average Darcy's friction factor in the main channel - being the substitutionary coefficient of resistance
for the cross-section of the main channel calculated for the wetted perimeter $P_m$ that includes lengths of the section dividing
lines ($P_m = P_l + P_{lsb} + P_b + P_{rsb} + P_r$), $R_m$ symbolizes the hydraulic radius of the entire cross-section of the main channel ($R_m = A_m/P_m$), and $R_i$ is the hydraulic radius of the cross-sectional area per given roughness ($R_i = A_i/P_i$). The coefficient of resistance
$f_m$ in the cross-section of the main channel is calculated on the basis of the average velocity ($v_m = Q_m/A_m$) and the calculated
hydraulic radius of the main channel $R_m$. The determined areas of the cross-sectional area $A_i$, in which the flow conditions are
shaped under the influence of a constant roughness over the length of the wetted $P_i$ perimeter, were used to calculate the
hydraulic radius $R_i$ and the coefficients of resistance $f_i$.





In Fig. 7, the calculated values of apparent resistance coefficients $f_a$ (subscripts $l$ and $r$ - the left and the right side, respectively)

and resistance coefficients of the main channel $f_m$ as well as of the bottom $f_{mb}$, the side slope of the main channel $f_{ms}$, the bottom

of the floodplain $f_{fb}$, of the compound cross section in experiments made in variants W 1.0, W 2.0, W 2.T1, W 3.T2, W 3.0 and

W3.T2 are presented as a function of the depth ratio $(H\text{-}h)/H$.

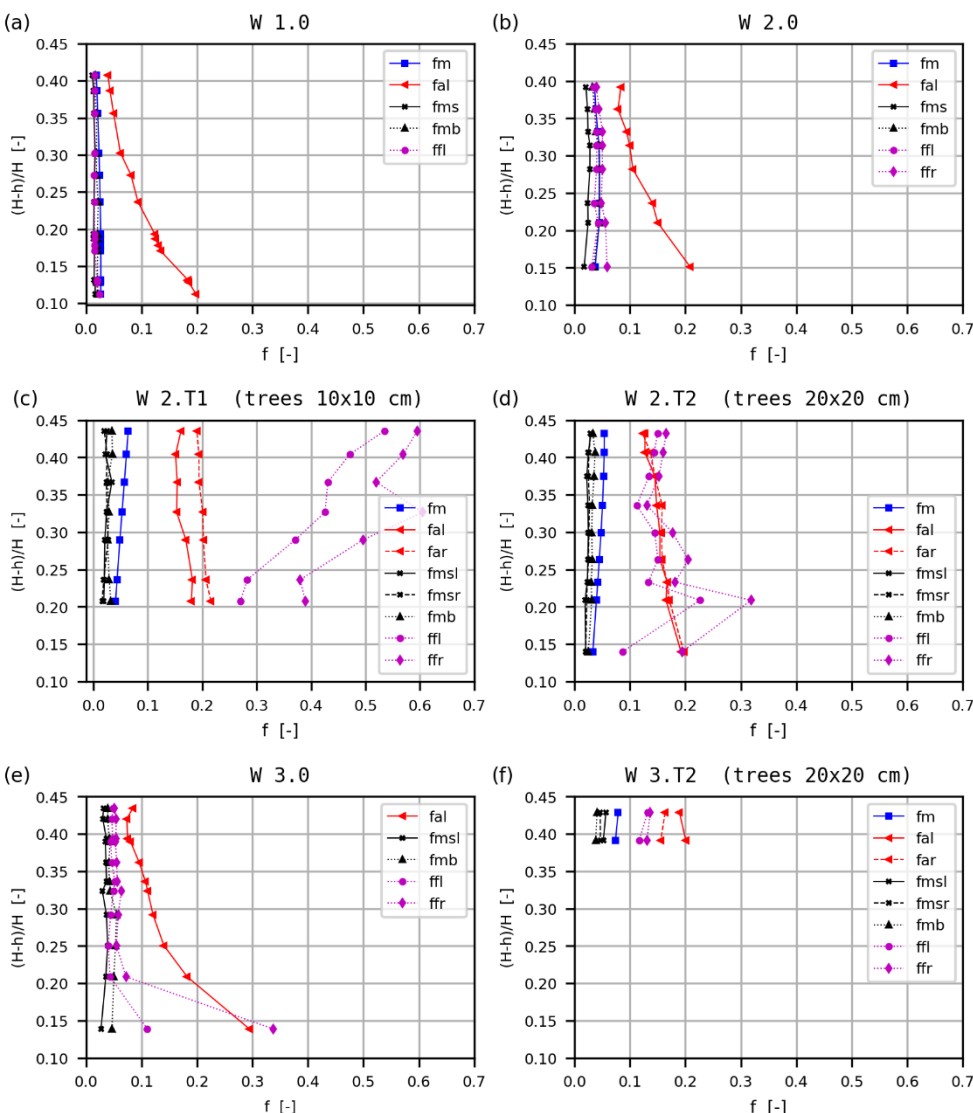

**Figure 7: Variability of resistance coefficients in the cross-section of the compound cross-section in variants: (a) W 1.0, (b) W 2.0, (c) W 2.T1, (d) W 2.T2, (e) W 3.0, and (f) W 3.T2.**

The values of the apparent resistance coefficients $f_a$ for the compound cross-section in variants W 1.0, W 2.0 and W 3.0, and

resistance coefficients of the bottom of the floodplain $f_{fb}$ with high roughness (in variants W 2.0 and W 3.0) decrease with the





increase of the flow depth (also with increase of the ratio $(H\text{-}h)/H$) (Fig. 7). In contrast, values of resistance coefficients for the
entire main channel $f_m$, $f_{ms}$ slopes and bottom of the main channel $f_{mb}$ in all variants of the study without trees and the floodplain
$f_{fb}$ in variant W 1.0, do not change significantly with depth. The presence of trees and the interaction process in variants W
2.T1, W 2.T2 and W 3.T2, in contrast to variants W 2.0 and 3.0, contributes to the fact that there is a significant increase of
apparent resistance coefficients values above the depth of $(H\text{-}h)/H = 0.2$.

Figure 8 presents the influence of floodplain trees on values of resistance coefficients $f_m$ in the main channel and apparent
resistance coefficients $f_a$ at the boundary between the main channel and floodplain in the second and third variants. The increase
in roughness on the floodplains and interactions resulted in an increase in the drag coefficient $f_m$ in the smooth main channel
in variant W 2.0 over the entire analyzed range (Fig. 8a). The coefficient values would first increase above $(H\text{-}h)/H = 0.25$ and
then start to decrease. At approximately $H = 0.25$ m ($(H\text{-}h)/H = 0.4$), the value of the $f_m$ coefficient in variant W 2.0 doubled
(by about 108 %) in the smooth main channel, which resulted in a 38.6 % reduction in flow in the entire channel (Table 1).
Additionally, trees from floodplains intensify the increase in resistance and subsequent reduction in flow. In variant W 2.T1
(trees 10x10 cm), the $f_m$ value increased by approximately 48 % in the main channel and the flow reduction was approximately
38.2 % in the entire channel (Table 1). In variant W 2.T2, with a larger spacing (20x20 cm) and fewer trees, the $f_m$ value
increased by approximately 36 % in the smooth main channel and the flow reduction was 24.1 % in the entire channel. The
values of the resistance coefficients $f_m$ in variants with trees (W 2.T1 and W 2.T2) increase with the increase of the flow depth
(Fig. 8a). The increase in the roughness of the main channel slopes in variant W 3.0 resulted in a significant increase in the $f_m$
value in the main channel (Fig. 8b), for $H = 0.251$ m by approximately 32 % ($(H\text{-}h)/H = 0.36$) and for $H = 0.262$ m by
approximately 36 % ($(H\text{-}h)/H = 0.39$), while the flow reduction in the entire channel was approximately 12.9 % and 15 %,
respectively. Figure 8b presents the influence of floodplain trees (20x20 cm) on values of resistance coefficients $f_m$ in the main
channel in the third variant. In variant W 3.T2, the $f_m$ value in the main channel increased by approximately 57 % for $H = 0.263$
m ($(H\text{-}h)/H = 0.39$) and for $H = 0.28$ m ($(H\text{-}h)/H = 0.43$) by approximately 70 %, while the flow reduction in the entire channel
was approximately 27.4 % and 31 %, respectively (Table 1). The increase in roughness on the main channel and the interaction
resulted in a very similar increase in the apparent resistance coefficients $f_{al}$ in the main channel in variant W 2.0 at the analyzed
flow depth (Fig. 8c). In variants with trees (W 2.T1 and W 2.T2), the $f_a$ values increase compared to the variant without trees
and also decrease with increasing depth. In both cases, the increase in the $f_a$ value is the smallest at low depth in the catchment
area, and increases with depth, which is the result of the increase in interactions between the main channel and the tree-covered
floodplain. At the depth of $H = 0.20$ m ($(H\text{-}h)/H = 0.21$) in variant W 2.T1 in the left division plane, the $f_{al}$ coefficient value
increased by 20 % (Fig. 8c). In the right division plane, the $f_{ar}$ coefficient value is even higher than the left one by 24 %, which
indicates asymmetric flow in the main channel, higher flow velocities on the left side of the main channel (Fig. 4, W 2.T1). It
is similar at the entire depth, but the variation in the coefficient value increases. At the depth of $H = 0.26$ m ($(H\text{-}h)/H = 0.39$)
in the left division plane, the $f_{al}$ coefficient value in variant W 2.T1 increased by about 81 % and the $f_{ar}$ coefficient value is
even higher than the left one by 52 %. In variant W2.T2, with a larger spacing (20x20 cm) and fewer trees, the $f_a$ values are
similar on the left and right sides of the main channel, and at the depth of $H = 0.20$ m ($(H\text{-}h)/H = 0.21$), the $f_a$ coefficient value





increased by about 12 % and at $H = 0.26$ m $((H-h)/H = 0.39)$ by about 58 % (Fig. 8c). The increase in the roughness of the main channel slopes in variant W3.0 caused the greatest increase in the $f_{al}$ value at small flow depths in the division planes

(about 41 %, $H = 0.18$ m, $(H-h)/H = 0.14$) and decreased with increasing depth (about 22 %, $H = 0.25$ m, $(H-h)/H = 0.36$) until the values become equal (Fig. 8d). Trees from floodplains in variant W 3.T2 (20x20 cm) at greater depths $((H-h)/H \geq 0.39)$ resulted in a more than two-fold increase in the $f_a$ value. In the left division plane, the $f_a$ value increased by 168 % at $H = 0.26$ m $((H-h)/H = 0.39)$ and by 125 % at $H = 0.28$ m $((H-h)/H = 0.43)$, while in the right division plane by 101 % and 178 % respectively.


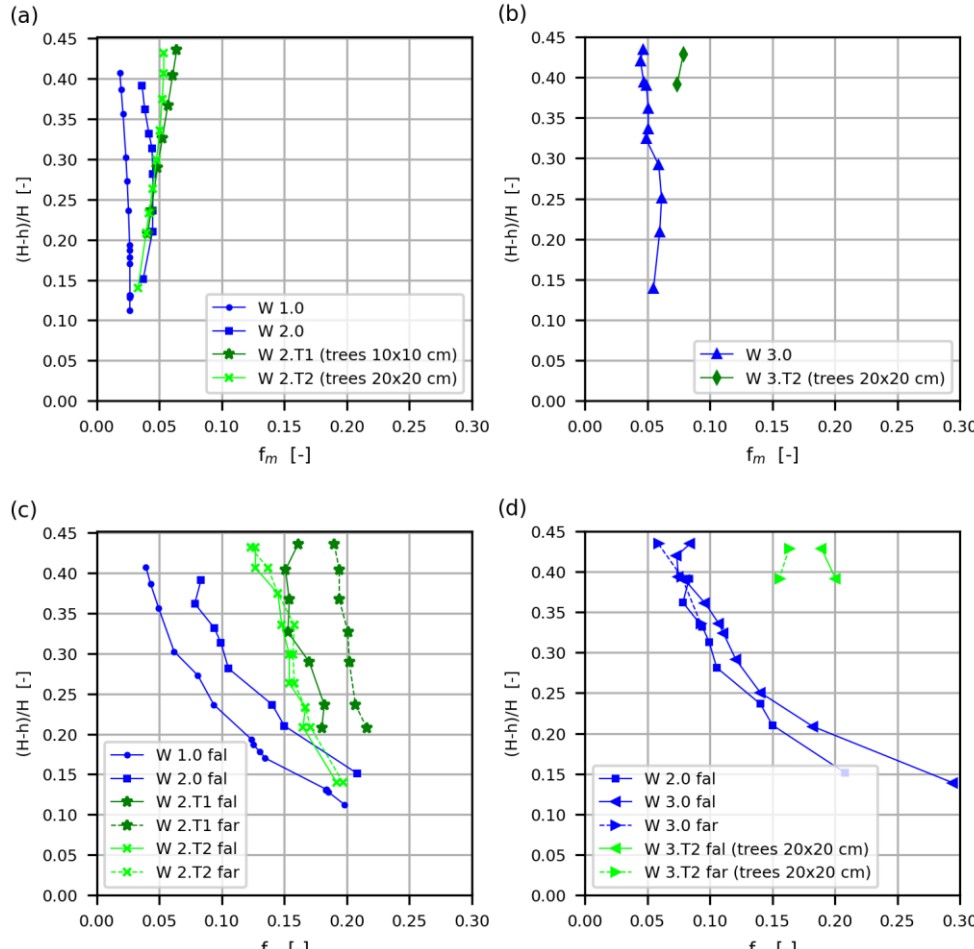

**Figure 8: Variability of resistance coefficients $f_m$ in the main channel and apparent resistance coefficients $f_a$ at the boundary between the main channel and the floodplain as a function of the flow depth.**

An increase in the flow depth in the floodplain resulted in an increase in the influence of the floodplain trees on the flow

conditions in the main channel and on the values of the apparent resistance coefficients $f_a$ at the apparent boundary of the main





channel and the floodplain (Fig. 8). However, at small flow depths ($(H\text{-}h)/H < 0.2$), the bottom roughness generally determines the coefficient values.

Figure 9 presents the influence of floodplain trees on values of resistance coefficients for the bottom $f_{mb}$ and the side slopes $f_{ms}$ of the main channel in the second and third tests. The $f_{mb}$ coefficient values did not change at the flow depth in the smooth
main channel (W 1.0, Fig. 9a), while with the increase in depth, the values initially increase slightly and then decrease in the channel with rough surfaces (W2.0 and W3.0, Fig. 9a-b). The increase on the floodplain roughness in W2.0 resulted in an even two-fold increase in the coefficient value (Fig. 9a). The influence of trees in the W2.T1 and W2.T2 variants resulted in the greatest decrease in the value of the $f_{mb}$ coefficient at small flow depths on the floodplain. As the flow depth increases, the value of the coefficient increases and already at approximately $(H\text{-}h)/H = 0.39$ the value is the same as in the variant without
trees.

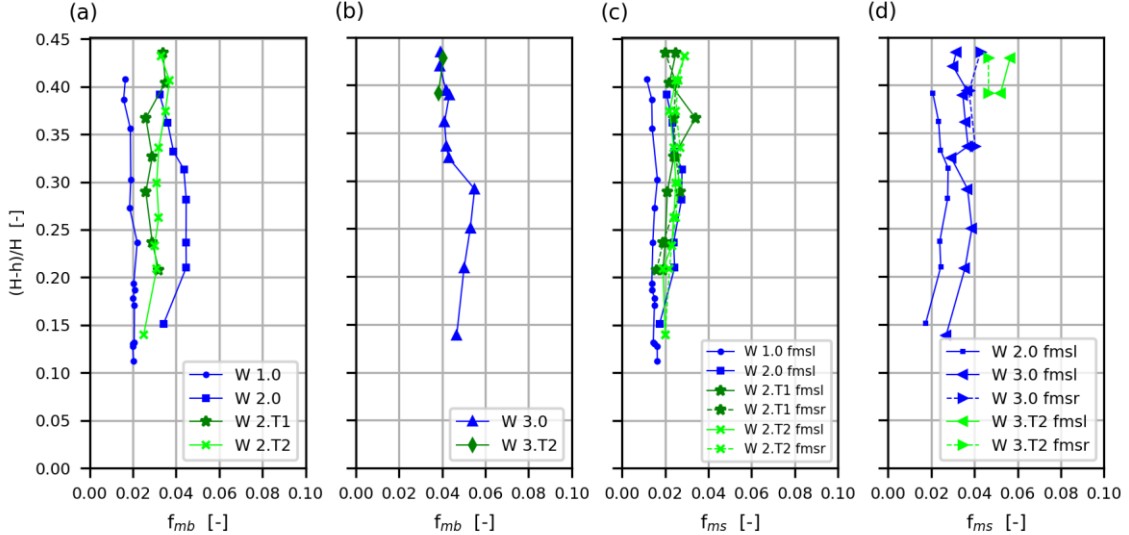

**Figure 9: Variability of resistance coefficients for the bottom $f_{mb}$ and the side slopes $f_{ms}$ of the main channel as a function of the flow depth.**


Figure 9b shows that the increase in the roughness of the main channel side slopes in W3.0 variant resulted in a slight increase in the $f_{mb}$ value. However, the floodplain trees in variant W3.T2 (20x20 cm) at higher flow depths ($(H\text{-}h)/H = 0.39\text{-}0.43$) did not result in a change in the $f_{mb}$ value for the bottom of the main channel. The increase in the surface roughness of the floodplains (W2.0, Fig. 9c) and the sloping banks of the main channel (W3.0, Fig. 9d) resulted in an increase in the $f_{ms}$ value
coefficient, also with an increase in the flow depth. The influence of trees in the smooth main channel (W2.T1 and W2.T2) result in only a slight decrease in $f_{ms}$ values at small flow depths in the floodplain, while at higher depths the coefficients did





not change. The influence of trees in the main channel with rough sloping banks (W3.T2) result in different increase in $f_{msl}$ and $f_{msr}$ values at higher flow depths (($H$-$h$)/$H$ = 0.39-0.43).

The influence of trees on the flow in a compound channel is very significant and the most common observed effect is a large
decrease in the water flow value (Table 1) and clear changes in the distribution of the depth average velocity in the cross-section of the compound channel (Fig. 10). It can be observed from the presented results that the influence of trees changes the values of resistance coefficients in the main channel to varying degrees, and the size of the changes depends on the surface roughness. Generally, the influence of trees in the smooth main channel resulted in a large increase in the apparent resistance coefficient but a slight decrease in the value of the bottom resistance coefficient with an almost unchanged resistance
coefficient of the main channel side slopes. However, the influence of trees in the channel with the rough surface of the main channel side slopes also resulted in a large increase in the apparent resistance coefficient and a small increase in the value of the resistance coefficient of the main channel side slopes with an unchanged bottom resistance coefficient at large flow depths (($H$-$h$)/$H$ = 0.39-0.43). The value of the apparent resistance coefficient depends on the surface roughness of the channel bottom, the density of trees and the depth of flow. Changes in the values of the apparent resistance coefficients can be explained by a
change in the interaction between the parts of the compound channel, which depends on the magnitude of changes in the average depth velocity in the main channel and on the floodplain (Fig. 10). If the values of the apparent resistance coefficients are all the greater, the greater are the differences between the flow velocities in the main channel and the floodplain.

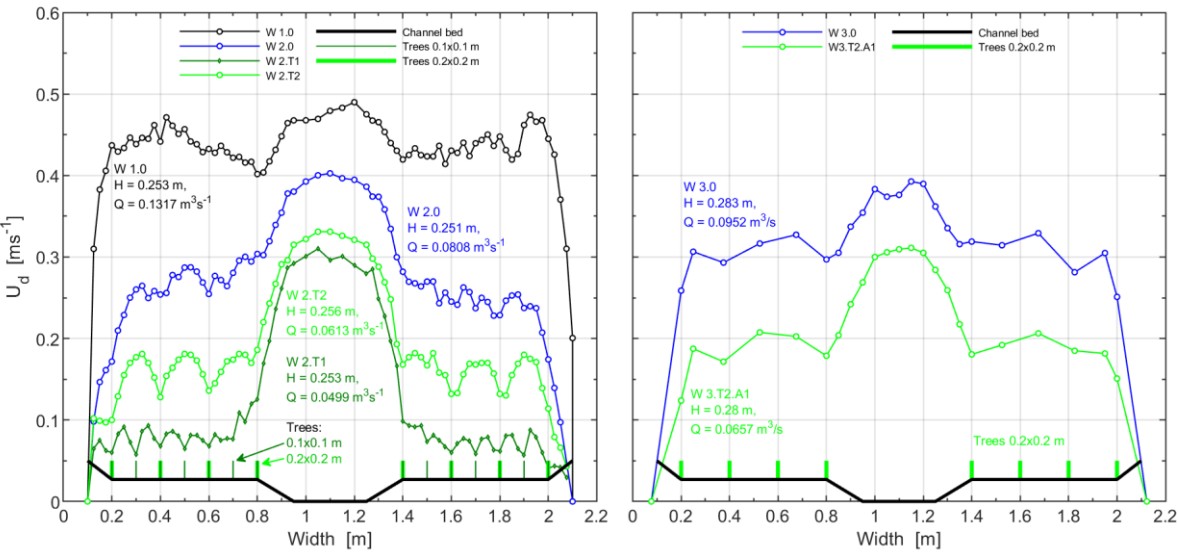

**Figure 10: Distribution of average velocity in verticals in variants W 1.0, W 2.0 and W 3.0 with similar flow depths in the compound channel.**





## 4 Conclusions

The analysis of the values of resistance coefficients determined for the main channel in the compound channel with different
bottom roughness with and without trees on floodplains showed that:

1) The increase in roughness on the floodplains (W 2.0) resulted in an increase in the interaction between the floodplains and
the main channel and, at the same time, an identical increase in the value of the apparent resistance coefficient for the
analyzed flow depths. The values of the apparent resistance coefficient differ little or are identical at very small flow depths
in floodplains in variants without and with trees in floodplains (the rough surface of the channel bottom generally has more
influence than trees). As the flow depth increased, the trees resulted in a significant increase in the value of the apparent
resistance coefficient and reached even twice as high values.

2) The increase in the roughness of the sloping banks of the main channel (W3.0) resulted in an increase in the value of the
apparent resistance coefficient for small flow depths, and only a slight increase in the value for larger flow depths. Trees in
the channel with rough floodplains and sloping banks of the main channel, at large flow depths resulted in a more than two-
fold increase in the value of the apparent resistance coefficient.

3) The values of the apparent resistance coefficients decrease with increasing flow depth, slower with trees on floodplains and
faster without trees.

4) In the smooth main channel and with rough floodplains the values of the flow resistance coefficient increase with the flow
depth up to $(H\text{-}h)/H = 0.25$, and then the values decrease. The floodplains trees resulted in a continuous increase in the value
of the flow resistance coefficient with the flow depth.

5) The values of apparent resistance coefficients are several times greater than the resistance coefficients for side slopes and
bottoms of the main channel. The floodplains trees in the smooth main channel resulted in a decrease the value of the
resistance coefficients for bottoms of the main channel below depth $(H\text{-}h)/H < 0.4$ and only a slight decrease the value for
side slopes of the main channel.

## Author contributions

AK, AKi and MK: conceptualisation. AK, AKi, MK, EK, JK, GM and MB: methodology and investigation. AK, AKi and
MK: writing – original draft. AK, AKi, MK, GM and MB measuring and analysis of data.

## Competing interests

The contact author has declared that none of the authors has any competing interests.



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
