# Peer review of "Effect of Floodplain Trees on Apparent Friction Coefficient in Straight Compound Channels"

_Hydrology and Earth System Sciences, 2024_

## Author Response (AR1)

Dear Editor and Reviewers

We would like to thank the reviewers for their careful review and insightful comments. We particularly appreciate the recognition of the article's topic being of interest to the journal. We fully agree with the reviewers' comments, especially regarding the need to clearly distinguish this manuscript from our previous work (Kubrak et al., 2019). We have ensured that the revised article minimizes overlap and emphasizes the novel aspects of the current study on the effect of tree configurations.

The first three chapters have been significantly changed or rewritten. The first variant without trees was almost completely removed. However, variants 2.0 and 3.0 had to stay because they need to be used as references to show the impact of trees. The bibliography has been supplemented, in fact there have been other articles on this topic, and some of them contain reviews on friction and compound channels that should be cited. All changes were marked in the manuscript using a color.

As suggested by the first reviewer, we changed the title of the article to:

"Effect of floodplain trees on apparent friction coefficient in straight compound channels".

As for the second reviewer's main comments:

to 1-11) almost everything has been included.

to 12) *"It would be useful to add images from the laboratory experiments. If dye has been added, the exchange processes could be at least discussed. This is mentioned in the abstract but the paper does not include results."*

> The photograph for the one of variants (/variant W3.T1/.) was included in the manuscript. For the /variant W3.T2/ the photograph was given in our previous work and we provided the reference. What about the dye: The results of instantaneous velocity measurements used as part of research on flow turbulence in a laboratory flume with trees did not include additional research using dyes. The dye was only used in laboratory studies of interactions between the main channel and floodplains, but not trees. The dye was used in other studies in a natural canal with and without vegetation (Kalinowska, M. B., Västilä, K., Nones, M., Kiczko, A., Karamuz, E., Brandyk, A., Kozioł, A., and Krukowski, M .: Influence of vegetation maintenance on flow and mixing: case study comparing fully cut with high-coverage conditions, Hydrol. Earth Syst., 27, 953–968, https://doi.org/10.5194/hess-27- 953-2023, 2023.). These studies were used to analyze the impact of river vegetation on various flow parameters.)

to 13) *"I recommend summarizing which average resistance factors result for the different setups (with and without trees). Are these values suitable to estimate the discharge? It would be interesting to show the determined vs. measured discharge for a given scenario."*

> The article was only intended to present how significant the influence of trees is on the values of individual coefficients. The authors are aware that the amount of data available from other studies does not allow providing "average" drag coefficients for different configurations (with and without trees). Even if so, to a very limited extent.

to 14) The applications have been rewritten.

Once again, we would like to thank you for your comments on our article. We believe that they allowed us to improve the manuscript significantly.

Best regards

Authors

---

## Author Response (AR2)

Dear Editors and Reviewers

We would like to thank the reviewers for their careful consideration and insightful comments. We particularly appreciate the recognition that the topic of the article is of interest to the journal.

All recent technical corrections suggested have been incorporated

The corrections made are shown below:

1) *Figure captions: Please check all figure captions. There are still references to removed subfigures (for example to W 1.0 and W 2.0 in Fig. 3 or to W 1.0 in Fig. 4).* Figure captions corrected

2) Suggested typos on lines 16, 20, 161 and 166 have been fixed

We would like to thank you again for your comments on our article.

Best regards

Authors